# Low dose rate γ-irradiation protects fruit fly chromosomes from double strand breaks and telomere fusions by reducing the esi-RNA biogenesis factor *Loquacious*

A. Porrazzo[1,2], F. Cipressa [2,3,4], A. De Gregorio[1], C. De Pittà[5], G. Sales [5], L. Ciapponi [1], P. Morciano[6], G. Esposito[7,8], M. A. Tabocchini [8] & G. Cenci [1,2✉]

It is still continuously debated whether the low-dose/dose-rate (LDR) of ionizing radiation represents a hazard for humans. Model organisms, such as fruit flies, are considered valuable systems to reveal insights into this issue. We found that, in wild-type *Drosophila melanogaster* larval neuroblasts, the frequency of Chromosome Breaks (CBs), induced by acute γ-irradiation, is considerably reduced when flies are previously exposed to a protracted dose of 0.4 Gy delivered at a dose rate of 2.5 mGy/h. This indicates that this exposure, which is associated with an increased expression of DNA damage response proteins, induces a radioadaptive response (RAR) that protects Drosophila from extensive DNA damage. Interestingly, the same exposure reduces the frequency of telomere fusions (TFs) from Drosophila telomere capping mutants suggesting that the LDR can generally promote a protective response on chromatin sites that are recognized as DNA breaks. Deep RNA sequencing revealed that RAR is associated with a reduced expression of *Loquacious D (Loqs-RD)* gene that encodes a well-conserved dsRNA binding protein required for esiRNAs biogenesis. Remarkably, loss of *Loqs* mimics the LDR-mediated chromosome protection as it decreases the IR-induced CBs and TFs frequency. Thus, our molecular characterization of RAR identifies Loqs as a key factor in the cellular response to LDR and in the epigenetic routes involved in radioresistance.

[1] Dipartimento di Biologia e Biotecnologie "C. Darwin", Sapienza Università di Roma, Rome, Italy. [2] Fondazione Cenci Bolognetti/ Istituto Pasteur Italia, Rome, Italy. [3] Centro Studi e Ricerche "Enrico Fermi", Rome, Italy. [4] Department of Ecological and Biological Sciences, Università degli Studi della Tuscia, Viterbo, Italy. [5] Dipartimento di Biologia, Università di Padova, Padua, Italy. [6] INFN-Laboratori Nazionali del Gran Sasso, 67100 Assergi, Italy. [7] Istituto Superiore di Sanita' ISS, Rome, Italy. [8] INFN-Roma 1, Rome, Italy. ✉email: giovanni.cenci@uniroma1.it

Low doses and low dose rates of ionizing radiation constitute one of various types of genotoxic stresses to which all living organisms are continuously exposed during their daily life. Human exposure to low dose/dose rates radiation depends on both natural (i.e., radon gas, terrestrial radiation and cosmic rays that penetrate the Earth's atmosphere) and man-made (i.e., such as X-rays, radiation used to diagnose diseases and for cancer therapy and fallout from nuclear explosives testing) sources. Low doses are defined as doses lower or equal to 0.1 Gy and low dose rates are defined as dose rates lower or equal to 0.1 mGy/min (that is 6 mGy/h) for low Linear Energy Transfer (LET) radiation[1]. Whereas there is no doubt that intermediate and high doses of ionizing radiation, delivered either acutely or during a prolonged period, have deleterious effects in humans, including, but not exclusively, cancer and developmental defects, the biological consequences at lower doses (below 100 mGy) are less clear[2,3]. Yet, the comprehension of low doses radiation risks has societal importance considering the concerns arising from occupational radiation exposure, screening tests for cancer, nuclear power generation, frequent-flyer risks, missions in space, and radiological terrorism. Several efforts have been made to evaluate the effects of low doses/low dose rate radiation from epidemiological studies. However, these studies are influenced by several uncertainties that include, but not limited to the radiation quality, the age at exposure, the effects of both internal and external exposures, gender and temporal factors[2,3]. Thus, to better assess the accuracy of risk estimates and increase the statistical power, the low dose/low dose rate epidemiological studies would require a large population to be analyzed or pooling of data from several observations.

The linear-no-threshold (LNT) model has long been used for setting dose limitations on radiation exposure, that is, basically assuming that all radiation exposures, regardless of how low the dose is, increase the risk for cancer[4]. The justification of using this model is that radiation carcinogenesis is triggered by DNA damage. Yet, the possibility of improving the LNT risk extrapolation model has been recently debated as other potential mechanisms affecting radiation carcinogenesis have been described[5–7]. These include epigenetic mechanisms, transmissible genome instability, bystander effects, radiation hormesis and Radio Adaptive Response[3,8].

The Radio Adaptive Response (RAR) is a protective phenomenon where a small initial low radiation dose (priming) reduces the response to a subsequent high radiation dose (challenging). The main proposed mechanisms to explain RAR are an increase of the efficiency of DNA repair activity and of the level of antioxidant enzymes[9–11]. First discovered in cultured human lymphocytes[12], RAR has been observed both in vivo and in vitro in several mammalian systems as well as in the zebrafish embryo model using various end points such as cell lethality, chromosomal aberrations, mutation induction and DNA repair[9,10]. Although the general view posits that RAR may be due to enhanced repair of DNA damage in the "primed" cells (those having been exposed to a priming dose) when compared to the "unprimed" cells, several experiments using different cell types and irradiation procedures produced conflicting results[13]. Thus, our understanding on the molecular mechanisms that underlie RAR is still elusive. Indeed, the manifestation of the response depends on several factors such as the cell, tissue, animal types, genetic background, p53 status. Moreover, the dose, dose rate and time period between priming and challenging dose may be crucial for a cell to induce RAR. It is still unclear what dose and dose rate might trigger RAR in humans[3,11]. This is also complicated by the high degree of inter-individual variation, which in turn depends on the radio-sensitivity of an individual. Thus, the evaluation of whether low radiation doses or dose rates represent a risk estimate factor for radiation protection purposes requires a better understanding of the RAR.

Recently, a link between radiation protection and telomere maintenance has emerged. The examination of leukocytes in atomic bomb survivors of Hiroshima revealed a long-term detrimental effect of an inverse correlation between telomere length and IR dose[14]. Effects on telomere length have also been observed during space missions as cosmic radiation has been shown to elongate telomeres. Interestingly, two days on the Earth restored normal telomere length indicating that elongation of telomeres can be considered as a well-established adaptive response to high radiation levels in space[15,16]. Nevertheless, exposure to high radiation background does not affect telomere length suggesting the existence of a well-defined threshold under which the effects of radiation exposure are undetectable[17,18].

Here, we show that Drosophila larvae that are chronically exposed to a specific priming γ-radiation dose of 0.4 Gy delivered at 2.5 mGy/h (0.4 Gy LDR) during embryo-to-third instar larvae development, exhibit a strong reduction of chromosome break frequency after a 10 Gy γ ray challenging dose, with respect to non-pretreated flies. Moreover, we demonstrate that the same LDR treatment reduces the frequency of telomeric fusions (TFs) associated with the loss of telomere capping suggesting that the 0.4 Gy LDR-mediated protective effect can be generally extended to chromatin sites that are also inappropriately recognized as DNA breaks. Deep RNA sequencing indicates that the 0.4 Gy LDR-mediated protective effect on chromosome breaks is associated with a reduction of the Loquacious D (Loqs-PD) isoform, a well conserved dsRNA binding protein required for the esiRNA biogenesis. Interestingly, loss of Loqs PD (but not of other isoforms) reduces IR-induced DSBs and TFs indicating that modulation of Loqs PD may represent an escape route for the cell to preserve chromosome integrity upon induction of genotoxic stress.

## Results

### A 2.5 mGy/h dose rate gamma irradiation protects Drosophila chromosomes from ionizing radiation-induced chromosome damage.
To investigate the effects of low dose rate irradiation on chromosome damage, we assessed whether a pretreatment with chronic low doses of γ rays (priming dose) affects the frequency of total chromosome breaks (CBs; for examples of CBs see Fig. 1a) induced by an acute γ dose of 10 Gy (challenging dose) in Drosophila third instar larvae neuroblasts. Vials containing 12 h old Drosophila embryos were placed at 23 °C inside the LIBIS irradiation facility[19] and irradiated continuously at a dose rate of 2.5 mGy/h until they developed into third instar larvae (7 days for a final dose of 0.4 Gy, herein 0.4 Gy LDR). Irradiated and non-irradiated control vials were then exposed to a challenging dose, at the dose rate of about 0.7 Gy/min, and 4 h later larval neuroblasts were analyzed for CBs frequency (Supplementary Fig. 1). Chromosomes were fixed at 4 h and 8 h post irradiation time points (PIR) to recover cells that were irradiated in the S-G$_2$ or G$_1$ stage, respectively[20]. The average number of total chromosome breaks per cell (CBs/cell) was calculated by measuring the ratio of the total number of chromatid deletions (CDs, scored as a single event) and isochromosome breaks (ISOs or chromosome deletions, scored as two events) to the total number of metaphases (Fig. 1b). Cytological analysis of exposed pretreated and non-pretreated third instar larvae showed that in non-pretreated larvae the average number of CBs/cell induced by the challenging dose after 4 h PIR was ~1.0 in females and ~0.8 in males. Differently, the 0.4 Gy LDR determined a ~2 fold decrease of CBs/cell frequency in both male and female larval neuroblasts, indicating that the LDR treatment renders neuroblast chromosomes resistant to IR-induced DNA breakage as a result of the activation

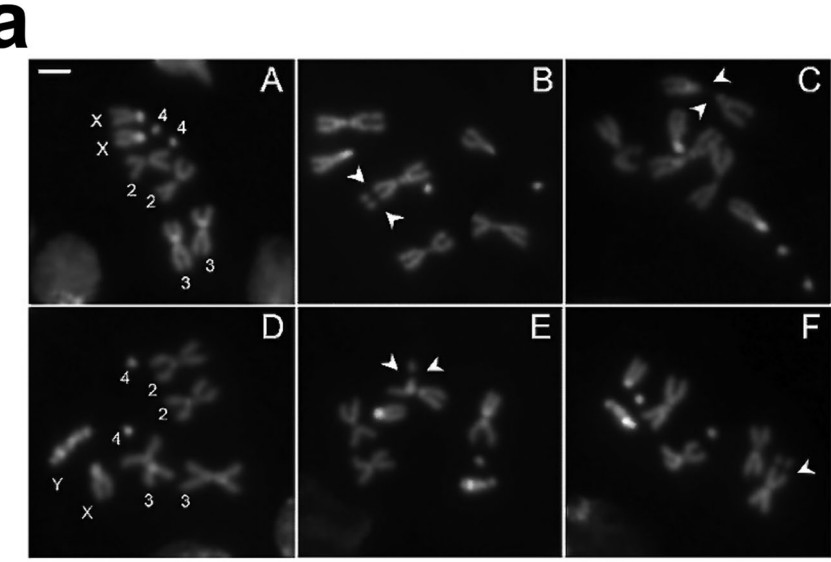

**Fig. 1 Effects of 0.4 Gy LDR treatment on CBs. a** Examples of IR-induced CBs in neuroblasts from *Oregon*-R third instar larvae. (A) Wild-type female and (D) male metaphases; (B) female and (C, F) male metaphases with examples of isochromosome breaks (arrows); (E) male metaphase showing autosomal chromatid deletions (arrows). Bar = 5 μm. **b** Analysis of the frequency of CBs induced by the acute exposure to 10 Gy, in 0.4 Gy LDR chronically treated and non-treated (NT) larvae after 4 and 8 h post irradiation (PIR), and in the first and the second generation from adults exposed to 0.4 Gy LDR during their development. Chromosome exchanges did not vary and were not considered in the analysis. The CBs frequency of 0.4 Gy LDR + 10 Gy treated larvae at 8 h and 4 h PIR was compared with that of 10 Gy treated larvae at the same PIR times. The CBs frequency of 10 Gy treated F1 and F2 was compared with that of 10 Gy exposed controls after 4 h PIR. Note that CBs frequency of 0.4 Gy LDR treated larvae were indistinguishable from that of NT larvae. $n = 3$ biologically independent experiments were conducted. LDR low dose rate, ISOs: isochromosome deletions, CDs chromatid deletions, CBs chromosome breaks, PIR post-irradiation time, F1 first filial generation, F2 second filial generation. All comparisons were performed using Student's *t*-test, †statistically significant for $p < 0.05$, ††statistically significant for $p < 0.01$.

of radioadaptive response (RAR). Interestingly, a 1.2 mGy/h dose rate for 7 days (0.2 Gy total dose) was not sufficient to trigger a RAR, suggesting that this response could be dose rate dependent (Supplementary Fig. 2a).

We sought to understand whether the 0.4 Gy LDR could affect CBs frequency also at 8 h PIR, when cells are irradiated during the $G_1$ stage. The analysis of ~1000 female metaphases revealed no statistically significant difference on CBs frequency between non-pre-treated (~54%) and 0.4 Gy LDR treated (~45%) larvae at 8 h PIR (Fig. 1b). This suggests that 0.4 Gy LDR could primarily influence the HR-based repair systems that, by employing sister chromatids as template for the DNA synthesis to regenerate the sequence surrounding the break site, is restricted to S-$G_2$ phase[21]. Furthermore, we evaluated whether the radiation resistance could

be also maintained for more than one generation. Thus, male and female adult flies that developed from 0.4 Gy LDR-pretreated larvae were mated to each other outside the LIBIS and the resulting $F_1$ larvae irradiated with 10 Gy. Interestingly, we found that the frequency of CBs, calculated at 4 h PIR, was ~2 fold reduced in metaphases from both male and female $F_1$ larvae of 0.4 Gy LDR -derived adult parents ($n = 1861$ for both sexes) with respect to control $F_1$ larvae deriving from non-pretreated adults (Fig. 1b). This indicates that the RAR shows a transgenerational inheritance. However, the $F_1$ obtained from reciprocal crossings between 0.4 Gy LDR-treated females or males and non-treated males or females, respectively, did not elicit RAR (Supplementary Table 1), suggesting that the transgenerational inheritance of RAR depends on both maternal and paternal factors.

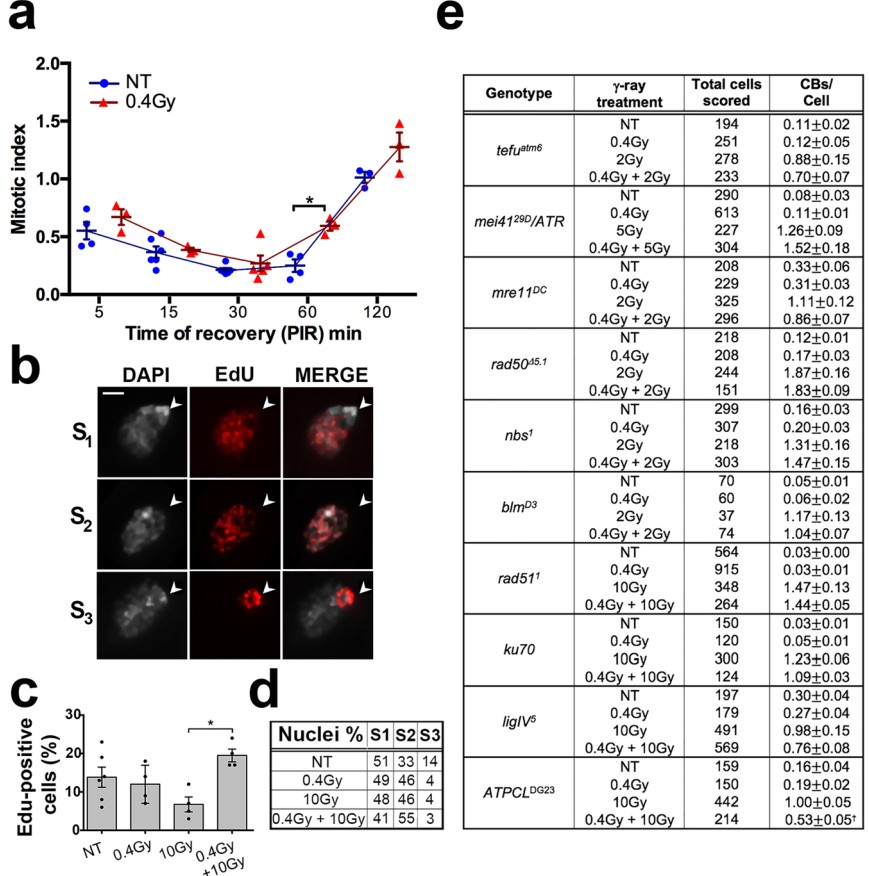

**e**

| Genotype | γ-ray treatment | Total cells scored | CBs/ Cell |
|---|---|---|---|
| tefu[atm6] | NT | 194 | 0.11±0.02 |
| | 0.4Gy | 251 | 0.12±0.05 |
| | 2Gy | 278 | 0.88±0.15 |
| | 0.4Gy + 2Gy | 233 | 0.70±0.07 |
| mei41[29D]/ATR | NT | 290 | 0.08±0.03 |
| | 0.4Gy | 613 | 0.11±0.01 |
| | 5Gy | 227 | 1.26±0.09 |
| | 0.4Gy + 5Gy | 304 | 1.52±0.18 |
| mre11[DC] | NT | 208 | 0.33±0.06 |
| | 0.4Gy | 229 | 0.31±0.03 |
| | 2Gy | 325 | 1.11±0.12 |
| | 0.4Gy + 2Gy | 296 | 0.86±0.07 |
| rad50[45.1] | NT | 218 | 0.12±0.01 |
| | 0.4Gy | 208 | 0.17±0.03 |
| | 2Gy | 244 | 1.87±0.16 |
| | 0.4Gy + 2Gy | 151 | 1.83±0.09 |
| nbs[1] | NT | 299 | 0.16±0.03 |
| | 0.4Gy | 307 | 0.20±0.03 |
| | 2Gy | 218 | 1.31±0.16 |
| | 0.4Gy + 2Gy | 303 | 1.47±0.15 |
| blm[D3] | NT | 70 | 0.05±0.01 |
| | 0.4Gy | 60 | 0.06±0.02 |
| | 2Gy | 37 | 1.17±0.13 |
| | 0.4Gy + 2Gy | 74 | 1.04±0.07 |
| rad51[1] | NT | 564 | 0.03±0.00 |
| | 0.4Gy | 915 | 0.03±0.01 |
| | 10Gy | 348 | 1.47±0.13 |
| | 0.4Gy + 10Gy | 264 | 1.44±0.05 |
| ku70 | NT | 150 | 0.03±0.01 |
| | 0.4Gy | 120 | 0.05±0.01 |
| | 10Gy | 300 | 1.23±0.06 |
| | 0.4Gy + 10Gy | 124 | 1.09±0.03 |
| ligIV[5] | NT | 197 | 0.30±0.04 |
| | 0.4Gy | 179 | 0.27±0.04 |
| | 10Gy | 491 | 0.98±0.15 |
| | 0.4Gy + 10Gy | 569 | 0.76±0.08 |
| ATPCL[DG23] | NT | 159 | 0.16±0.04 |
| | 0.4Gy | 150 | 0.19±0.02 |
| | 10Gy | 442 | 1.00±0.05 |
| | 0.4Gy + 10Gy | 214 | 0.53±0.05† |

**Fig. 2 The influence of cell cycle and DNA Damage Response (DDR) on the RAR. a** Checkpoint assay of *Oregon*-R 0.4 Gy LDR chronically exposed *Oregon*-R larval brains versus non-treated (NT) controls. In the picture is shown the mitotic index (MI) calculated as the ratio between the total number of dividing cells and the number of optical fields. Note that the MI of 0.4 Gy LDR larvae increases after 60 min post-irradiation compared to NT control. $n = 3$ biologically independent experiments were conducted. Values of $p < 0.05$ (*) were considered as statistically significant (Student *t*-test). **b** EdU incorporation analysis of *Oregon*-R larval brain cells exposed to IR. EdU-positive cell nuclei of larval brains show 3 different incorporation patterns: $S_1$ represents early S phase nuclei that do not incorporate Edu in the DAPI-stained chromocenter; $S_2$ represents mid-S phase nuclei that incorporate EdU in all DAPI-stained nuclei; $S_3$ shows nuclei in the late S-phase that incorporate EdU only in the chromocenter. **c** Frequencies of EdU positive nuclei. The frequencies of EdU positive nuclei in each sample (NT, 0.4 Gy LDR, 10 Gy, 0.4 Gy LDR + 10 Gy) were obtained by examining at least 2000 nuclei from 3 brains. Values of $p < 0.05$ (*) were considered as statistically significant (Student *t*-test). **d** Quantification (%) of EdU positive cells divided in S1, S2, and S3 phases according to the staining pattern observed in (b). No statistical differences were observed in treated cells compared to controls. **e** DNA repair gene mutant larvae do not elicit RAR. Frequency of CBs induced by an acute 2 Gy, 5 Gy or 10 Gy irradiation on 0.4 Gy LDR Drosophila strains mutated in genes involved in the DNA damage response (DDR). The *ATPCL* mutant was used as a control. Note that in all DNA repair (but not in *ATPCL[DG23]*) mutants RAR is abolished. ISOs isochromosome deletions, CDs chromatid deletions, CBs chromosome breaks, NT non-treated, PIR post-irradiation time. The comparisons of the CBs frequency were performed among 0.4 Gy LDR treated- and untreated mutants before and after the exposure to the challenging dose using Student's *t*-test; †statistically significant for $p < 0.01$. Bar = 10 µm.

To analyze whether the 0.4 Gy LDR affected the regulation of $G_2/M$ checkpoint, we carried out a standard checkpoint assay evaluating the proportion of dividing cells (Mitotic Index, MI) in larval brain cells fixed at different times after the challenging dose (10 Gy). As shown in Fig. 2a, the MI of the non-pretreated Oregon R control brains dropped at 5–30 min PIR, remained low for 1 h and came back to a normal value only at 2 h PIR. The MI of 0.4 Gy LDR -pretreated larvae decreased as much as in control brains at 5–30 min PIR, but then markedly increased after 30 min PIR, suggesting that the 0.4 Gy LDR exposure determined a faster recovery from the $G_2$-M checkpoint (Fig. 2a) compared to non-pretreated cells, most likely due to a reduced amount of DNA damage (~50%) to be repaired.

We also checked whether the rate of IR-induced apoptosis and levels of reactive oxygen species (ROS) were influenced by the 0.4 Gy LDR. Immunostaining of whole-mounted brains and imaginal discs for cleaved-Caspase-3 revealed that the number of positive cells in treated larval brains was indistinguishable from that observed in non-pretreated control after the challenging dose, indicating that RAR does not reduce the occurrence and distribution of apoptosis caused by extensive DNA damage (Supplementary Fig. 3a-d). Moreover, using dihydroethidium (DHE) as an indicator we found that ROS levels in 10Gy-irradiated brains did not differ from those observed in 0.4 Gy LDR + 10Gy-irradiated brains, although, as expected, they increased compared to unirradiated larvae (Supplementary Fig. 4). These results indicate that LDR does not lead to a statistically significant response to the ROS generating activity that could be taken into account to explain the radioresistance.

We finally compared the occurrence of DNA replication in non-pretreated and pretreated brain cells by analyzing the incorporation of the EdU (5-ethynyl-2'-deoxyuridine) analog of thymidine, before and after IR (see Materials and Methods). Before 10 Gy irradiation, non-pretreated wild-type and 0.4 Gy

LDR-pretreated larvae, exhibited a similar proportion of nuclei (~13%; $n = $ ~2000 cells) that were actively replicating their DNA and incorporated EdU. However, after irradiation, whereas in non-pretreated wild-type brains the frequency of EdU-labeled nuclei dropped to ~6.5% (as a consequence of the arrest of S phase), 0.4 Gy LDR-pretreated brains displayed a proportion of positive nuclei (~20%; $n = $ ~2000 cells) very similar to uni-rradiated larvae (Fig. 2c). Based on the EdU incorporation pattern, according to Cenci et al., 2015[22], we subdivided the S phase in three subclasses, namely $S_1$ (early/mid S; nucleus partially or completely stained with the exception of the chromocenter), $S_2$ (mid/late S; staining at the chromocenter and at a less compacted nuclear area) and $S_3$ (late S; only chromocenter stained; Fig. 2b). This classification revealed us that pretreated and non-pretreated cells did not differ in the relative frequencies of nuclei showing $S_1$, $S_2$ and $S_3$ EdU incorporation patterns (Fig. 2d). Altogether, these results indicate that 0.4 Gy LDR-pretreated cells recover much faster than non-pretreated controls from the DNA damage-induced replication block.

**The RAR to chromosome aberrations relies on a more efficient DNA damage response.** To determine whether the reduction of chromosome breaks promoted by 0.4 Gy LDR depends on the presence of proteins required for specific DSB repair pathways, we checked whether 0.4 Gy LDR-pretreated larvae bearing mutations in genes encoding DSB repair master proteins (dATM, dRad50, dNbs1, dMre11), HR-specific repair proteins required for resection or homology search and strand invasion (dRad51, Bloom), and NHEJ-specific proteins (ligase IV, Ku70) elicited a decrease of CBs frequency after the challenging dose. The analysis of mitotic chromosomes from all mutants revealed no significant difference ($p < 0.05$) in the frequency of CBs between 0.4 Gy LDR-pretreated and non-pretreated larvae, indicating that the 0.4 Gy LDR-mediated resistance to γ-ray-induced CBs is strictly dependent on DNA repair genes (Fig. 2e).

We then verified whether 0.4 Gy LDR affects the kinetics of γH2Av recruitment to chromatin breaks. H2Av is the functional homolog of H2AX in Drosophila and its phosphorylation at Ser147 is induced by DSBs[23]. 0.4 Gy LDR -pretreated and non-pretreated larvae were exposed to 10 Gy γ rays and then checked by IF for the presence of γH2Av foci in larval neuroblasts (Fig. 3a) after 5 min, 15 min, 30 min and 60 min PIR. We found that in non-pretreated control larval brains, the number of γH2Av foci/cell progressively increased over the time, reached the highest peak (6 foci/cell) at 30 min PIR and started to slowly decrease at 60 min PIR (Fig. 3b). LDR -pretreated neuroblasts also showed a progressive increment of the γH2Av foci/cell frequency over the time with a peak at 30 min PIR. Interestingly, the average of foci/cell at 30 min was around ~4 indicating that, differently from non-pretreated controls, LDR-pretreated brains accumulate less damage, most likely as the results of either an accelerated DNA damage response or a more efficient DNA repair (Fig. 3b). We have also analyzed the kinetics of γH2Av accumulation by WB and found that, consistently with our IF results, at 30 min PIR γH2Av in 0.4 Gy LDR-pretreated cells does not accumulate as robustly as in non-pretreated cells (Fig. 3c,d; see Supplementary Fig. 5 for unprocessed blot images), confirming that 0.4 Gy LDR could accelerate the completion of the DNA damage response. We also analyzed the levels of the MRN complex as well as of Ku70 in both pretreated and non-pretreated cell extracts. WB analyses revealed a statistically significant increase of Rad50 and Nbs1, but not of Ku70, protein levels in 0.4 Gy LDR-pretreated cells with respect to non-pretreated cells (Fig. 3e, f; see Supplementary Fig. 6 for unprocessed blot images). Collectively these data, along with the genetic analyses indicating

a strict requirement of DNA repair genes, suggest that the RAR relies on an efficient DNA damage response. However, we cannot rule out the possibility that the LDR could also influence chromatin compaction making it more reluctant to damage or more accessible to DNA repair factors compared to non-pretreated chromatin.

**LDR protects chromosomes from telomere dysfunction.** We also asked whether other DNA damage-induced events are also affected by LDR. We focused on DNA damage response elicited by dysfunctional telomeres and verified the occurrence of telomere fusions (TFs) in mutants with a defective telomere capping. Depletion of either terminin complex components and/or terminin-associated factors induces the formation of multicentric linear and ring chromosomes very likely because of NHEJ-mediated fusion events that occur at uncapped telomeres[24,25] (for examples of TFs see Fig. 4a). We thus measured the frequency of TFs of 0.4 Gy LDR-pretreated mutants in selected genes coding for telomere capping proteins, such as *verrocchio (ver)* and *caravaggio (cav)* that specify the terminin proteins VER and HP1-ORC-Associated Protein (HOAP), or *Su(var)205, pendolino (peo)* and *effete (eff)* that encode the non terminin protein Heterochromatin Protein 1a (HP1), Peo and UbcD1, respectively[24,25]. As indicated in Fig. 4b, we found that in all mutants analyzed, 0.4 Gy LDR yielded a relevant reduction of TFs indicating that this exposure exerts a protective effect at uncapped chromosome ends. We have also analyzed the frequency of TFs caused by mutations in *tefu, nbs, rad50* and *mre11* genes encoding for the DNA repair factors ATM, Nbs, Rad50 and Mre11, respectively and which are also required for protecting telomeres from fusion events[26,27]. Interestingly, we found that the frequency of TFs in DNA repair mutants was indistinguishable from that of non-pretreated mutants (Fig. 4b) indicating that, similarly to RAR, the 0.4 Gy LDR-induced telomere protection depends on DNA repair factors. However, the reduction of TFs did not always correlate with a reduction of γH2Av-positive Telomere Induced Foci (TIF) indicating that, unlike chromosomal DSBs, the reduction of fusigenic feature or uncapped telomeres does not have a direct effect on the cytological detection of telomere damage (Supplementary Fig. 7a, b). We finally checked if LDR could induce an increase of retro-transposition of the telomeric retroelement HeT-A, as it occurred in flies that were chronically exposed to sub-lethal doses of paraquat[28]. RT-qPCR analysis revealed no differences in the HeT-A expression between non-pretreated and 0.4 Gy LDR treated TF mutants (Supplementary Fig. 7c), indicating that the 0.4 Gy LDR protective effect on uncapped telomeres is unlikely due to an adaptive role of telomere elongation against the loss of capping.

**Differential gene expression and genetic analyses reveal loquacious (Loqs) as a RAR responsive factor.** To assess whether the 0.4 Gy LDR RAR was accompanied by differential modulation of specific set of genes, we performed a genome wide RNA-seq analysis of total RNAs extracted from unirradiated (NT), 10Gy-, 0.4 Gy LDR- and 0.4 Gy LDR + 10Gy-irradiated larval male brains. We decided to focus our transcriptomic analyses on males to avoid the confounding effects of gene expression variability of females which is known to be affected by dietary conditions during development and associated to investment into reproduction-related processes[29,30]. To search for gene functions specifically associated to the chromosome break phenotype, RNAs of 10Gy- and 0.4 Gy LDR + 10 Gy irradiated larvae were extracted from brains dissected after 4 h following the acute dose exposure. Pairwise comparisons revealed that the 0.4 Gy LDR treatment per se has not determined a statistically significant

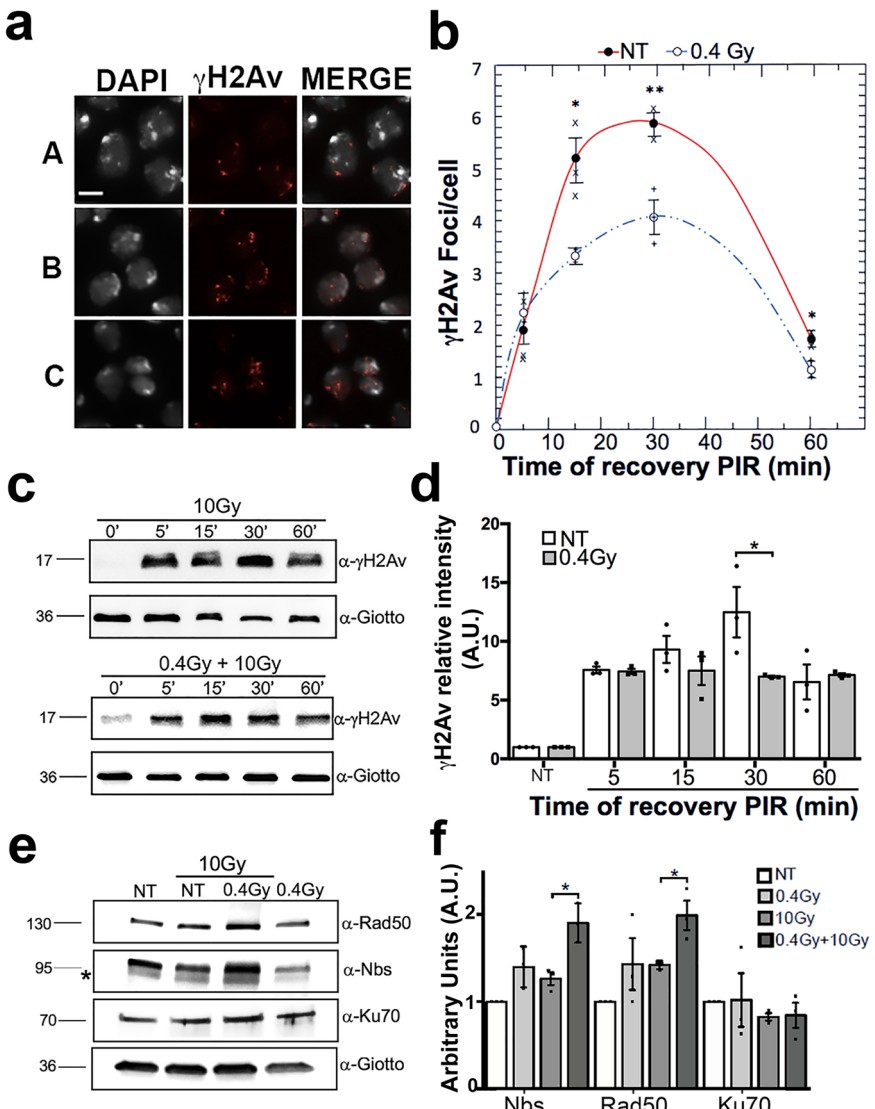

**Fig. 3 DNA damage accumulation in 0.4 Gy LDR -treated and non-treated larval brains. a** Anti -γH2Av immunofluorescence on 10 Gy irradiated cells showing (A) low damage (0–3 γH2Av spots per cell); (B) medium damage (4–7 γH2Av spots per cell); (C) high damage (>7 γH2Av spots per cell **b** Quantification of γH2Av foci/cell at different PIR time points; non-pretreated (close circles) and 0.4 Gy LDR -pretreated (open circles). Error bars represent the standard errors of the mean; lines only represent guide for the eye. Values of $p < 0.01$ (**) and $p < 0.05$ (*) were considered as statistically significant (Student $t$-test). **c** Western Blot analysis of γH2Av levels in treated and NT brains at 5 min, 15 min, 30 min, and 60 min PIR after acute IR exposure of 10 Gy. Giotto (VIB) was used as a loading control. **d** Quantification of γH2Av signals from the WB analysis. WBs from three independent experiments were used for the quantification. (*$p < 0.05$; Student $t$-test); PIR post-irradiation times. **e** Western Blot analysis of Nbs, Rad50, and Ku70 levels in treated and non-treated brains after 4 h from acute IR exposure of 10 Gy. Giotto (VIB) was used as a loading control. **f** Quantification of Nbs, Rad50, and Ku70 signals from the WB analysis. NT non-treated. Bar = 10 μm.

modulation of general levels of transcripts compared to the unirradiated condition. In contrast, both 10Gy- and 0.4 Gy LDR + 10Gy-treatments yielded a moderate, yet statistically significant, modulation of transcription in all comparisons (see Supplementary Data 1–5 for a list of both upregulated and downregulated genes). The Venn diagram including all differentially expressed genes (DEGs) from 0.4 Gy LDR + 10 Gy vs. 10 Gy, 0.4 Gy LDR + 10 Gy vs. 0.4 Gy LDR and 10 Gy vs. 0.4 Gy LDR comparisons, has shown only a small number of genes in common among all comparisons (Fig. 5a). To further identify genes potentially involved in the RAR, we focused on the set of DEGs found in the 0.4 Gy LDR + 10 Gy *vs.* 10 Gy comparison. DAVID functional enrichment analysis (EASE score < 0.05, Adjusted $p < 0.05$) revealed that the DEGs from this comparison are involved in RNA processing (see Heat Map of Fig. 5b;

Supplementary Data 6). 3 genes (*l(3)72Ab, CG5205* and *CG8064*) were also differentially expressed in the 0.4 Gy LDR + 10 Gy vs 0.4 Gy LDR comparison.

Among these unique genes, we have drawn our attention to the modulation of *loqs (loquacious)* transcripts that encode the dsRNA binding protein Loqs, which is required for siRNA and micro-RNA biogenesis and that is the fly homolog of mammal TRBP[31,32]. Four different transcripts have been identified for *loqs* (*loqs-RA, loqs-RB, loqs-RC* and *loqs-RD*) that give rise to the Loqs-PA, Loqs-PB, Loqs -PC and Loqs –PD isoforms, respectively. Loqs PA and Loqs PB both harbor three dsRBDs (L1, L2, L3) while Loqs PC and PD both lack the third dsRBD and instead have short aa stretches at their C-termini. Loqs-PA and Loqs-PB interact with Dcr-1 during miRNA biogenesis[32–35]. Loqs-PC is rarely expressed and has no known function[32] while Loqs-PD

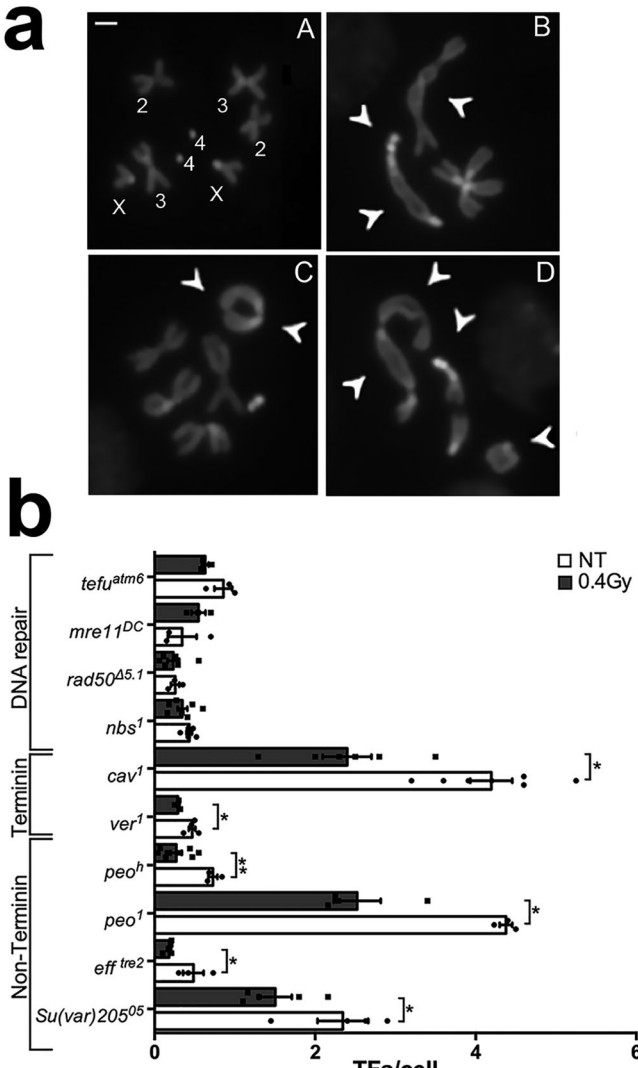

**Fig. 4 Analysis of TFs frequency in telomere capping and DDR mutants exposed to 0.4 Gy LDR. a** Examples of TFs scored for the TF frequency analysis. (A) Female wild-type metaphase; (B) female metaphase showing a Double Telomere Association (DTA) between the chromosomes 2 (arrows), a DTA involving both chromosomes 4 and both chromosomes X (arrows); (C) female metaphase showing a ring consisting of both X chromosomes (arrows) and a DTA between the chromosomes 4 (arrow); (D) female metaphase showing a multicentric chromosome configuration involving 3DTAs, a DTA between chromosome 4 and XR and a ring chromosome (arrows). **b** TF frequency (DTAs only) in 0.4 Gy LDR-pretreated and non-treated *Drosophila* strains mutated in genes encoding DNA damage repair factors (*tefu*, *mre11*, *rad50*, and *nbs*), proteins of terminin complex (*cav*, and *ver*), or proteins not associated to terminin complex (*peo*, *eff*, and *Su(var)205*). (*$p < 0.05$; **$p < 0.01$). DTAs double telomere associations; NT non-treated. Bar = 5 μm.

facilitates Dcr-2–dependent endosiRNA biogenesis[32,33]. Our RNA-Seq experiment showed that the *loqs-RD* (but not the other transcripts) appeared ~2 fold decreased in 0.4 Gy LDR + 10 Gy samples compared to 10 Gy irradiated samples. This reduction was also validated by RT-qPCR on three independent RNA extractions (Fig. 6a) suggesting that LDR could specifically influence Loqs-dependent esi-RNA biogenesis. Consistent with this, we found that 0.4 Gy LDR + 10 Gy larvae showed a statistically significant increase of *mus308* transcripts that are

the endogenous targets of *esi-1* RNA[36], with respect to both unirradiated and 10 Gy irradiated larvae (Fig. 6a).

To further understand whether the reduction of Loqs indeed accounts for the RAR, we analyzed chromosome integrity in chronically irradiated Loqs-depleted larval brain cells with and without the 0.4 Gy LDR-treatment. We thus measured the frequencies of CBs in the *loqs^KO* null mutant upon 0.4 Gy LDR, 10 Gy and 0.4 Gy LDR + 10 Gy treatments as well as in unirradiated *loqs^KO* brains using the same experimental set up described above. Intriguingly, we found that while unirradiated *loqs^KO* larval brain cells exhibited very rare CBs, the frequency of CBs in this mutant was >50% reduced with respect to Or-R control larvae ($n = 250$) after the exposure to 10 Gy (Fig. 6d). This suggests that the depletion of Loqs could mimic the effect of the 0.4 Gy LDR priming dose in determining resistance to ionizing radiation-induced CBs. Interestingly, we observed that this response was lost when Loqs-PD, but not –PB, was expressed in the *loqs^KO* null mutant (Fig. 6d), confirming that only the reduction of this specific isoform is involved in triggering radio-resistance to CBs. However, we observed that the >50% reduction of CB frequency in 10 Gy irradiated *loqs^KO* null mutant, did not further decrease upon the 0.4 Gy LDR pretreatment, suggesting that the effect of Loqs loss on the radio-resistance is epistatic over the 0.4 Gy (2.5mGy/h) LDR priming dose (Fig. 6d). Finally, we observed a ~50% reduction of CBs also in *loqs^KO/+* heterozygotes, suggesting that radioresistance effect is dominant and confirming our RNA-seq data that even a moderate downregulation of this gene can account for the protection against extensive DNA breakage.

To confirm that the loss of Loqs mirrors the effects of the 0.4 Gy LDR -induced radio-resistance, we first verified whether the response to CBs in Loqs-depleted cells is also dependent on DNA repair factors. We thus sought to analyze CB frequency in *loqs^KO; tefu^atm* and *loqs^KO; nbs^1* double mutants. While *loqs^KO; tefu^atm* double mutants turned out to be early lethal, thus preventing us for a further characterization, we found that the frequency of CBs in 2 Gy irradiated *loqs^KO; nbs^1* larvae was similar to that of 2 Gy irradiated *nbs^1* single mutants (Supplementary Table 2), confirming that the protection from chromosome breakage in Loqs-depleted and 0.4 Gy LDR cells requires shared DNA protein factors. Finally, we observed that the kinetics of γH2Av recruitment and the response to production of ROS in *loqs^KO* and in 0.4 Gy LDR + 10 Gy cells, were similar (Fig. 6e; Supplementary Fig. 4). Altogether, this set of evidences confirms that repression of Loqs explains the LDR-mediated RAR and endorses the view that Loqs-PD is a bona fide low dose responsive gene.

**Loss of Loqs reduces the frequency of Telomere Fusions in Drosophila telomere capping mutants**. We then sought to verify if a reduction of Loqs, similarly to LDR, affects the fusigenic ability of chromosomes with defective telomere capping. We thus checked the frequency of TFs in double mutant combinations between *loqs^KO* and TF mutants (*ver^1, cav^1, nbs^1*) mapping on the third chromosome and therefore more suitable for recombination with *loqs^KO*, located on the second chromosome. Very interestingly, we found that in *loqs^KO; ver^1* and *loqs^KO; cav^1* mutant combinations the percentage of TFs/cell was substantially lower than that observed in the corresponding controls, while, as expected, TF frequency of *nbs1* mutants was not affected by loss of Loqs (Fig. 6f). This finding further confirms that, similarly to RAR, the reduction of Loqs is also involved in the protection of uncapped telomeres.

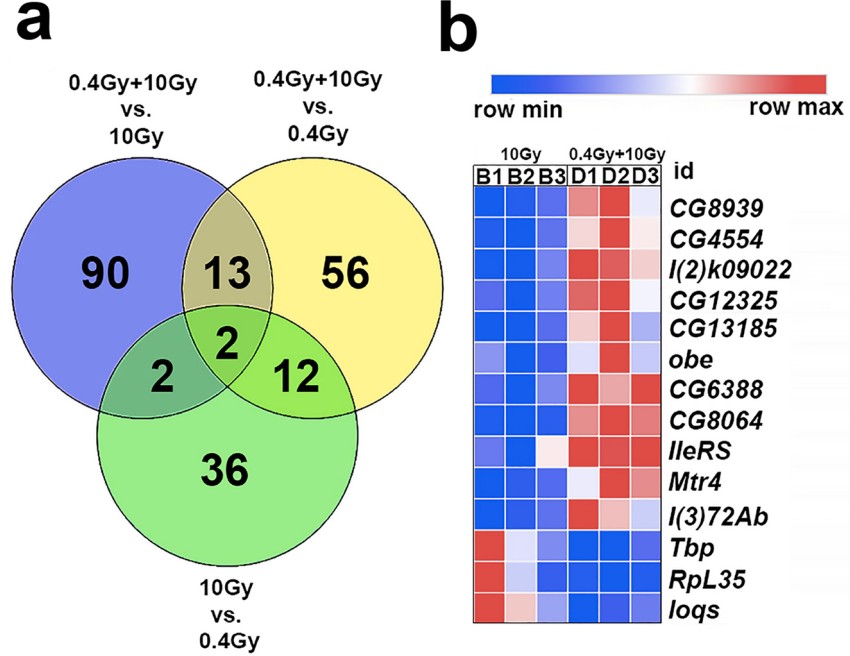

**Fig. 5 Analysis of differentially expressed genes in radio adaptive response. a** Representation of the Venn diagram including all differentially expressed genes from 0.4 Gy LDR + 10 Gy vs. 10 Gy, 0.4 Gy LDR + 10 Gy *vs.* 0.4 Gy LDR and 10 Gy *vs.* 0.4 Gy LDR comparisons. Only a very low percentage of genes were in common among all comparisons. A complete list of differentially expressed genes is provided in the Supplementary Data 2, 3 and 5. **b** Heat map representing the differentially expressed genes from 0.4 Gy LDR + 10 Gy *vs.* 10 Gy comparison belonging to "RNA processing" biological process obtained from DAVID functional enrichment analysis (Supplementary Data 6). The analysis was performed on three biological replicates obtained from three independent experiments, namely 1, 2, and 3 from the 10 Gy (B) and 0.4 Gy LDR + 10 Gy (D) conditions. A color-coded scale for the normalized expression values is used: red and blue represent high and low expression levels, respectively.

## Discussion

We showed that a protracted exposure to LDR protects Drosophila chromosomes from IR-induced DSBs in mitotic cells, by promoting a more efficient DNA damage response. Our genetic and cytological characterizations indicated that this response, dubbed RAR, is enhanced when DSBs occur after DNA replication and depends on both HR and NHEJ DNA repair pathways, which are normally activated during the S-G$_2$ checkpoint[37]. We also found that RAR is associated to increased levels of the Rad50 and Nbs proteins but not of Ku70, suggesting that the upregulation of MRN complex could play a pivotal role in the LDR-promoted DNA repair. However, it cannot be excluded that other components of either HR or NHEJ pathway could be differentially regulated as a consequence of RAR, a point that deserves further in-depth analyses as soon as more antibodies against Drosophila DNA repair factors will be made available. We also demonstrated that LDR reduces also the frequency of TFs of *Drosophila* capping mutants, revealing, to the best of our knowledge, an unprecedented protective effect on dysfunctional chromosome ends. Since Drosophila has emerged as a well-established model organism for human telomere biology[25,38], we can envisage that our findings could provide a valid counter-measure to inhibit telomere fusions in cancer cells. It can be argued that the reduction of TFs mediated by the 0.4 Gy LDR, which enhances DNA repair, could appear in contradiction with the evidence that telomeres normally avoid unwanted DNA repair[39]. However, it is also known that DNA repair proteins interact with telomere capping proteins to establish a protective structure during G$_2$-M[39–41]. We can thus hypothesize that the 0.4 Gy LDR could reinforce the DNA repair protein recruitment that has been impaired by the loss of capping factors, thus counteracting telomere dysfunction/fusions. The findings that some Drosophila capping proteins indeed interact with the MRN

complex[42,43] and that loss of either ATM or MRN induces TFs[38], gives further support to this hypothesis. Interestingly, the observation that the frequency of TFs associated to depletion of either ATM or MRN is not affected by 0.4 Gy LDR provides a robust evidence that, similarly to the RAR, the protective effect on uncapped telomeres is dependent on the activation of DNA-damage response. Thus, our results indicate that the protective effects of LDR on CBs and TFs share common molecular bases.

Our total RNA seq analyses revealed that mRNA profile of "primed" cells is undistinguishable from that of "unprimed" cells before the induction of DSBs indicating that the 0.4 Gy LDR per se has no effect on gene expression. However, we cannot exclude that LDR could change the expression of small non-coding RNAs, such as microRNAs. Furthermore, 0.4 Gy LDR does not seem to affect life parameters as 0.4 Gy LDR treated flies are fertile and do not exhibit changes in the life span (Supplementary Fig. 8). However, compared to unprimed cells, the induction of DSBs in primed cells yielded a discrete modulation of gene transcription, which might help understand the molecular bases underlying the LDR protective effects. Our evidence that the downregulation of *loqs-RD* transcript is required to prevent extensive chromosome breakage in primed cells could suggest a link between esiRNAs biogenesis and radio adaptive response. Remarkably, we found that depletion of *loqs-RD*, similarly to 0.4 Gy LDR, confers radio-resistance to mitotic chromosomes upon DSBs induction, confirming that the downregulation of this dsRNA binding protein can account for the 0.4 Gy LDR-mediated response to DNA damage. In light of this, we can certainly state that *loqs-RD* can be considered the first 0.4 Gy LDR-responsive gene ever characterized. In addition, we would like to point out that an understanding of the molecular basis of RAR could help identify unanticipated factors involved in radioresistance.

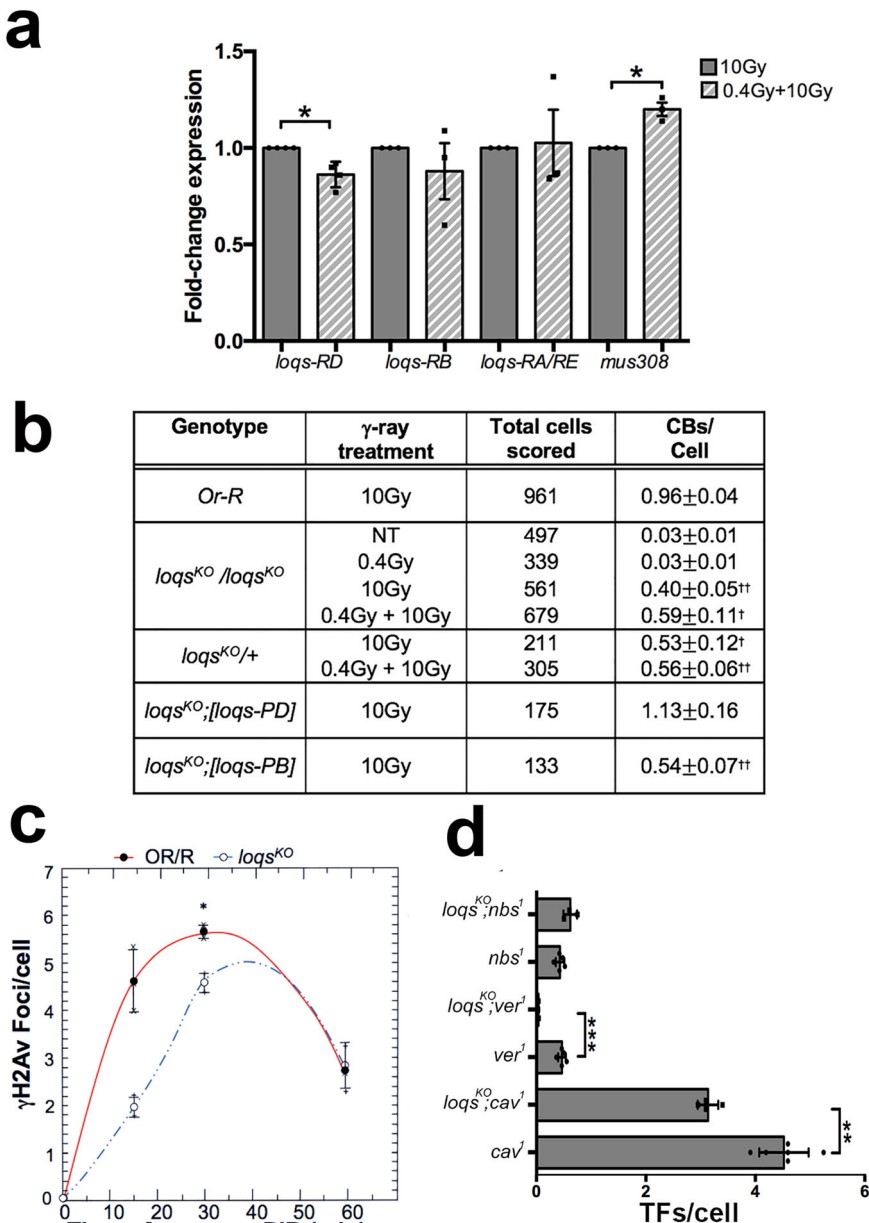

**Fig. 6 Loqs as a RAR responsive factor. a** RT-qPCR on different *Loqs* splicing variants and *mus308*. *Loqs-RD* (but not *-RB, -RA*, and *-RE*) transcripts are downregulated in 0.4 Gy LDR exposed male brains following the exposure to 10 Gy while *mus308* transcripts are upregulated. Three different experiments were performed. The statistical analysis used was Student *t*-test (*$p < 0.05$). **b** Frequency of IR-induced CBs in 0.4 Gy LDR treated and non-treated (NT) homozygous and heterozygous *loqs^KO* mutants, *Oregon*-R controls, and *loqs^KO* null mutants expressing the Loqs-PD and Loqs-PB exogenous isoforms. CBs frequencies of untreated and 0.4 Gy LDR treated *loqs^KO* mutants were compared each other. The remaining comparisons of CBs frequency were performed among untreated and 0.4 Gy LDR treated *loqs^KO* mutants exposed to 10 Gy and 10 Gy treated *OR-R*. CBs: chromosome breaks; †statistically significant for $p < 0.05$, ††statistically significant for $p < 0.01$ (Student's *t*-test). **c** Kinetics of γH2Av recruitment after irradiation with 10 Gy in *loqs^KO* mutants; lines only represent guide for the eye. Student *t*-test (*$p < 0.05$). **d** The graph shows the TFs revealed by counting only DTAs in *loqs;ver[1]*, *loqs;cav[1]* and *loqs;nbs[1]* double mutants as well in *ver, cav* and *nbs* single mutants. The statistical analysis was performed by Student's *t*-test (**$p < 0.01$; ***$p < 0.001$). DTAs double telomere associations.

Loqs-PD interacts with Dicer-2 through its C-terminal region, and is crucial for efficient production of subset of siRNAs, namely hairpin-derived endo-siRNA, cis-Natural Antisense Transcript (NAT)-derived endo-siRNA, as well as exo-siRNAs generated from an inverted repeat transgene[44]. However, Loqs-PD is not essential for the production of transposon-derived endo-siRNAs and virus-derived exo-siRNAs[45]. Locus-specific siRNA formation has been observed associated to DSBs in different systems, including flies[46–48] and promoting repair in mammalian cells and plants[47,49]. However, as the inactivation of siRNA pathways does not result in DNA repair defects in flies, the significance of siRNA formation in Drosophila DNA repair remains unclear[50]. Thus, it is plausible to consider that endogenous siRNAs derived from cis-antisense transcripts as well as from other dsRNA sources that are normally converted to siRNA by Loqs-PD, do not play a direct role in DNA repair thus explaining why *loqs* mutants do not elicit defects in DNA repair. It is also plausible that Loqs-PD could influence DNA repair independently of its involvement in siRNA

biogenesis or that the accumulation of esiRNA precursors could account for a more proficient DNA damage response. Yet, given the high level of homology between Loqs-PD and its human counterpart TARBP2/PACT, it would be extremely interesting to evaluate in the future whether TARBP2/PACT could also induce protective effects on human chromosomes.

## Methods

**Drosophila strains**. Mutants in the DNA repair genes $mei41^{29D}$, $rad50^{\Delta5.1}$, $mre11^{DC}$, $nbs^1$, $tefu^{atm6}$ and in telomere capping genes $cav^1$, $moi^1$, $ver^1$, $Su(var)$ $205^{05}$, $eff^{tre2}$, $peo^h$, and $peo^1$ were made available to us by the researchers who first characterized them[22,51–58]. The mutant stocks $lig4^5$ (Bl#8519), $rad51^1/SpnA1$ (Bl#3322), and $loqs^{KO}$ (Bl#65411) were obtained from the Bloomington Stock Center. The mutant stock ku70$^{EX8}$ was obtained from W. R. Engels (University of Wisconsin, Madison, WI). The $blm^{D3}$ stock was provided by M. Gatti (SAPIENZA University of Rome, Italy). The $loqs^{KO}$; $[loqs-PA]$, $loqs^{KO}$; $[loqs-PB]$ and $loqs^{KO}$; $[loqs-PD]$ lines were a generous gift from P. Zamore (University of Massachusetts Medical School, Worcester, MA). The Oregon R strain was used as a control in all experiments. All strains were maintained at 25 °C on Drosophila medium (Nutri-Fly®GF; Genesee Scientific) treated with propionic acid. The detailed information on the balancers and the genetic markers used are available online on Flybase (http://flybase.bio.indiana.edu/).

**Irradiation treatments**. All the irradiations were carried out at the Istituto Superiore di Sanità (ISS, Rome, Italy). For the chronic γ ray treatment 12 h embryos from 20 young females and 20 young males were irradiated at the LIBIS γ-irradiation facility with $^{137}$Cs sources[19]. The 0.4 Gy LDR and the 0.2 Gy priming doses were chronically delivered with a dose rate of 2.5 mGy/h and 1.2 mGy/h, respectively during embryo to third instar development (7 days at 25 °C). For the RAR analysis, chronically exposed third instar larvae were irradiated with 10, 5 and 2 Gy γ rays (challenging dose) from $^{137}$Cs sources, at a dose rate of 0.7 Gy/min using the Gammacell Exactor 40 (Nordion). Following irradiation, non-irradiated and irradiated flies were maintained in the same incubator at 25 °C.

**Chromosome cytology and microscopy**. To obtain DAPI (4,6-diamidino-2-phenylindole)-stained, colchicine-pretreated Drosophila larval brain chromosome preparations for the analysis of chromosome aberrations and telomere fusions, larval brains were dissected in NaCl 0.7% and treated for 1 h with $10^{-6}$ M colchicine[58]. Successively, they were transferred in a hypotonic solution (0.5% sodium citrate), and then they were fixed in 45% acid acetic, squashed on the slides, and immediately frozen in liquid nitrogen. After the removal of coverslips, the slides were stained with DAPI/VECTASHIELD® (VECTOR Laboratories). At least 150 metaphases for each condition were analyzed through direct observation using the inverted fluorescence microscope Nikon TE 2000 (Nikon Instruments Inc., Americas) equipped with a Charged-Coupled Device (CCD camera; Photometrics CoolSnap HQ).

**Immunostaining and γH2Av foci detection**. To evaluate the IR-induced γH2Av foci kinetics of recruitment to DNA damage sites, brains from irradiated third instar larvae were dissected and fixed at various post-irradiation times (PIR). The samples were incubated in 3.7% formaldehyde for 5 min, transferred for 30 s in 45% acetic acid, and fixed in 65% acetic acid. Successively, the brains were squashed and frozen in liquid nitrogen. After the removal of the coverslips, the slides were placed in cold ethanol for 15 min, rinsed for 15 min in 0.1%TritonX-100/PBS (PBT), and incubated with the anti-γH2Av primary antibody diluted in PBT (1:10; Mouse; Developmental Studies Hybridoma Bank, IA 52242) overnight at 4 °C. Then, all slides were incubated with the fluorescein (FITC)-conjugated AffiniPure Donkey Anti-Mouse IgG (H + L) (1:150; Jackson ImmunoResearch) for 1 h at room temperature (RT). Immunostained preparations were mounted in VECTA-SHIELD® Antifade Mounting Medium containing DAPI (4,6 diamidino-2-phenylindole) (VECTOR Laboratories). To quantify the foci at least 1000 cells were analyzed for each PIR. The slides were analyzed using a Zeiss Axioplan epifluorescence microscope (Carl Zeiss, Obezkochen, Germany) equipped with a cooled CCD camera (Photometrics, Woburn, MA, USA). The digital images in grayscale were acquired separately, converted in photoshop format, RGB stained, and merged.

**EdU incorporation and staining**. EdU labeling was performed as previously described[22] using the ClickiT Alexa Fluor 488 Imaging kit (Invitrogen). Brains of third instar larvae were treated with 10 μM EdU for 60 min and then fixed and analyzed as described above. To quantify the EdU-positive cells, at least 2000 cells were analyzed for each condition. For the statistical analysis, three different experiments were performed.

**Apoptosis levels analysis**. Wing imaginal discs and brains were dissected from third instar Oregon-R larvae in NaCl 0.7% after 8–10 h following the acute exposure with 10 Gy. The samples were placed in 4% formaldehyde diluted in PBS 1X for 20 min and then permeabilized in 1%Triton-100/PBS for 15 min. The samples were incubated with the anti-Cleaved Drosophila Dcp-1 (1:50; Rabbit; Asp216 #9578; Cell Signaling Technology®) overnight at 4 °C and the next day with the AlexaFluor® 488 AffiniPure Donkey Anti-Rabbit IgG (H + L) (1:300; Jackson ImmunoResearch) for 30 min. After a further wash of 15 min in PBT, the samples were stained with DAPI and mounted on the slides. The slides were analyzed using the inverted fluorescence microscope Nikon TE 2000 (Nikon Instruments Inc., Americas) equipped with a Charged-Coupled Device (CCD camera; Photometrics CoolSnap HQ). The digital images in grayscale were acquired separately, converted in photoshop format, RGB stained, and merged. The analysis of digital pictures was performed using ImageJ 1.8.0 (Image Processing and Analysis in Java).

**Reactive oxygen species (ROS) detection**. ROS were measured using the in vivo Dihydroethidium (DHE) staining adapting the protocol of ref.[59]. Briefly, third instar larval brains were dissected in Phosphate Buffer Saline (PBS 1X) within 15 min after irradiation treatments. Brains were incubated in 3 μM DHE (-CAS 38483-26-0-Calbiochem) for 5 min. After two washes in PBS 1X for 5 min, brains were gently fixed in 3,7% formaldehyde for 5 min, rapidly washed again, and immediately observed using an inverted fluorescence microscope (see below). Images were analyzed using the ImageJ software. To analyze the intensity of DHE fluorescence, the mean grey value was measured for each image.

**Antibody generation**. To obtain the anti-Ku70 antibody, rabbits were immunized with a 6×His-tagged C-terminal polypeptide of Ku70 encompassing amino acids 1–256 (UniProtKB Q23976 KU70_DROME). The resulting antisera were affinity-purified by standard methods. Rabbit immunization and antisera affinity purification were carried out by Agro-Bio (La Ferté St Aubin, France).

**Protein extracts and Western blotting**. Protein extracts from Drosophila larval brains were obtained by dissecting 10 larval brains in 0.7% NaCl and homogenizing them in 20 μl of 2X Laemmli buffer. Protein samples were loaded into a 4–20% Mini-PROTEAN TGX precast gel to perform electrophoresis (SDS-PAGE) and blotted using the Trans-Blot® Turbo™ Transfer System on a nitrocellulose membrane (Hybond ECL, Amersham). Filters were blocked in 5% non-fat dry milk dissolved in 0.1% Tween-20/PBS for 30 min at RT and, then, incubated with anti-Giotto (1:5000; Rabbit; ref.[60]) and anti-γH2Av (1:1000; Mouse; Developmental Studies Hybridoma Bank, IA 52242) overnight at 4 °C. The membranes were then incubated with HRP-conjugated anti-Mouse and anti-Rabbit IgGs secondary antibody (1:5000; Amersham) for 1 h at RT and then washed again 3 times with 0.1%Tween-20/PBS. The chemiluminescent signal was revealed through either SuperSignal™ West Femto or SuperSignal™ West Pico substrate (Thermo Scientific™) using the ChemiDoc scanning system (Bio-Rad). Band intensities were quantified by densitometric analysis using the Image Lab 4.0.1 software (Bio-Rad). WB was repeated independently at least three times.

**Genomic DNA extraction and RT-qPCR**. To evaluate Het-A abundance, DNA was extracted from third instar larvae by standard procedures. The amount of Het-A was assessed by RT-qPCR using the following primers:
Het-A_FW 5'-ACCATAATGCCAACAGCTCC-3'
Het-A_RV 5'-AGCCAGCATTGCAGGTAGTT-3'
The Reference gene used was RP49 (see below).

**Total RNA extraction and RNA seq**. Drosophila RNA was isolated from brain of third instar larvae exposed to four different irradiation treatments (non-pretreated control, 0.4 Gy chronic exposure, 10 Gy acute exposure, and 0.4 + 10 Gy). Brains (100 brains/sample) were dissected in triplicate and RNA extracted using TRIzol (TRI Reagent® SIGMA Life Science, Sigma-Aldrich). The RNA-Seq of each individual sample was carried out from IGA Technology Services (Udine, Italy). cDNA libraries were constructed with 1 μg of total RNA by using the "Ovation® Drosophila RNA-Seq System" (TECAN-NuGEN, Redwood City, CA, USA) following the manufacturer's instructions. The total RNA was fragmented 3 min at 94 °C and DNA contaminants removed with 1X Agencourt AMPure XP beads (Agencourt Bioscience Cooperation, Beckman Coulter, Beverly, MA, USA). Total RNA samples and final cDNA libraries were quantified with the Qubit 2.0 Fluorometer (Invitrogen, Carlsbad, CA, USA) and quality tested by Agilent 2100 Bioanalyzer Nano assay. For cluster generation on the flow cell, libraries were processed with cBot (Illumina, San Diego, CA, USA) following the manufacturer's instructions. Sequencing was carried out in single-end mode (1 × 75 bp) by using NextSeq500 (Illumina) with a targeted sequencing depth of about 40 million reads per sample. Raw data were processed with the software CASAVA v1.8.2 (Illumina) for both format conversion and de-multiplexing. Resulting reads were trimmed to remove adapter sequences using cutadapt (version 1.15) and then analyzed using the bcbio-nextgen pipeline (version 1.0.6). Specifically, reads were aligned against the Drosophila melanogaster reference genome (BDGP6) using STAR (v2.5.3a); gene abundances were estimated with Salmon (v0.9.1). Differential expression analysis at the gene level was performed using the DESeq2 R package. Genes with an adjusted p-value (FDR) < 0.05 after correction for multiple testing (Benjamini–Hochberg method) were considered differentially expressed. lncRNAs polyA+ were also

detected as the RNAseq library was polyA+ enriched. Finally, to analyze the functional relationships of these differentially expressed genes, a Gene Ontology (GO) functional enrichment analysis through the DAVID v. 6.8 web tool (https://david.ncifcrf.gov/, ref. [61]) was performed.

**cDNA amplification and qPCR**. To validate the expression levels of transcripts selected from the RNA seq analysis, equal amounts of cDNA were synthesized from 300 ng of total RNA for each sample by using the iScript™ cDNA Synthesis Kit (Bio-Rad, Hercules, CA, USA). Thirty nanograms of cDNA per reaction were analyzed for semi-qPCR using the SsoAdvanced™ Universal SYBR® Green Super-mix Kit (Bio-Rad) following the manufacturer's protocol. The thermal cycling conditions were: 50 °C (2 min), 95 °C (10 min) followed by 40 cycles at 95 °C (15 s), 60 °C (1 min), and 95 °C (15 s), 60 °C (1 min) 95 °C (15 s), and 60 °C (15 s). The specificity of the reaction was verified by melting curve analysis. The PCR primers used were:

loqsRD_FW 5'-GCAAGGGCAAAAGCAAGAAGA-3'
loqsRD_RV 5'-TTGAATGATACTCACTTCGCCCT-3'
loqsRB_FW 5'-ACTGCTTAAGTTACAGAAGA-3'
loqsRB_RV 5'-GGCCAGAGAAGGTCTTCTCC-3'
loqsRA/RE_FW 5'-GTCTGCAGGAGACTCCCATC-3'
loqsRA/RE_RV 5'-TCAAGCAAGTTTCGCCCTCC-3'
mus308_FW 5'-AGCGGACGACAAGGAGAATG-3'
mus308_RV 5'-GGAGAACTCGTCCCGGAAAA-3'
rp49 was amplified as a reference transcript using the following primers:
rp49_FW 5'-CCGCTTCAAGGGACAGTATCT-3'
rp49_RV 5'-ATCTCGCCGCAGTAAACGC-3'

PCR reactions were carried out in the ABI Prism 7300 System (Applied Biosystems, Foster City, CA, USA). Data processing was performed using the ABI SDS v2.1 software (Applied Biosystems). The critical threshold value was noted for each transcript and normalized to the internal control. The fold change was calculated using the comparative $2^{(-\Delta\Delta Ct)}$ method.

**Statistics and reproducibility**. Data were presented using the mean ± standard error (SE) obtained from at least three independent experiments. All the statistical analyses were performed with Sigmaplot 11.0 (Systat Software, Imc., Chicago, IL, USA). The comparison between the two groups was analyzed by the Student $t$-test. The results were considered statistically significant when the $p$ values were <0.05.

**Reporting summary**. Further information on research design is available in the Nature Research Reporting Summary linked to this article.

## Data availability

All the raw reads used for the analysis of differential expression have been deposited at the Sequence Read Archive (NCBI), under BioProject PRJNA747152. All supplementary materials, including the uncropped blot images, are available in the Supplementary Information and Supplementary Data files. The data that support the findings of this study are available from the corresponding author upon reasonable request.

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

## Acknowledgements

We are grateful to Prof. P.D Zamore (University of Massachusetts Medical School) for sharing all *loqs* transgenic flies. We thank P. Anello (ISS, Rome, Italy) for technical assistance with the γ irradiators. This work has been supported by grants from the Institute Pasteur of Paris (PTR-24-2017), and Institute Pasteur of Rome to GC, by grants from FERMI Institute for Multidisciplinary Studies (Cosmic Silence Project, Italy), INFN-CSN5 (RENOIR experiment) and ISS-INFN Operative Agreement for R&D activities in the field of Radiobiology to MAT.

## Author contributions

Conceptualization: A.P., A.T., G.E., G.C. Methodology: A.P., A.T., G.E., G.C. Validation: A.P., F.C., A.D.G., P.M., G.S., C.D.P. Formal Analysis: A.P.,G.E., C.D.P., G.S., L.C. Investigation: A.P., A.D.G., F.C., P.M., L.C. Data Curation: A.P., C.D.P., G.S., G.E. Supervision: A.T., G.C. Funding Acquisition: A.T., G.C. Writing: G.C.

## Competing interests

The authors declare no competing interests.
