## [Peer Review File · Communications Biology]

Reviewers' comments:

Reviewer #1 (Remarks to the Author):

Comments for Porrazzo et al.

In Radio Adaptive Response (RAR), an initial exposure of a low dose radiation protects the organism from a subsequent challenge of radiation at a much higher dosage. Although RAR has been known for a long time, the underlying mechanism remains obscure. In this study, the authors investigated the genetic control of RAR using the classic *Drosophila* model. Although most of the studies remains descriptive, the amount of cytological work, the number of mutations tested, and the variety of genetic pathways covered in this study are unprecedented. In addition, the authors' uncovering of *Loquacious*, a small RNA biogenesis factor, as a potential modulator of RAR adds new directions to future studies of RAR. I only have minor comments for the authors.

(1) Tables in Figures 1, 2, and 6: it would be helpful to state specifically the pair-wise comparison from which a p value was generated for as there are so many different combinations possible.

(2) Page 3, line 61, please define "LET".

(3) It would be helpful to further describe the difference between *Loq-D* and the other isoforms in terms of domain structures.

(4) Please make sure all comparisons are supported by robust statistics.

Reviewer #2 (Remarks to the Author):

The authors have examined the adaptation to low doses of gamma irradiation, given at a low rate (LDR). This can induce the phenomenon of the radio-adaptive response (RAR), a physiologic reaction that may help the organism to cope with chronic radiation exposure. This is an interesting topic and it is an open debate whether low doses of radiation received at a low rate are dangerous or not. The authors have used the model organism *Drosophila melanogaster* to obtain mechanistic insight into the RAR.

Specifically, they examined the number of cytologically detected chromosome breaks in mitotic larval brain neuroblasts upon exposure to a high dose of gamma radiation given at a high rate, the challenge dose. They demonstrate that certain low-dose treatments induce an RAR that can partially protect chromosomes from breaking during the challenge dose. Based on the timing of their analysis they conclude that this effect is restricted to the S-G2 phase (HR window). This effect showed a trans-generational inheritance for F1 animals where BOTH parents had been given the RAR-inducing treatment, while treating only the male or female parent did not result in a detectable effect in the F1. This observation suggests epigenetic effects, but this was not followed up with further detail. The authors thoroughly characterize the RAR effects by demonstrating differences in the checkpoint response duration / cell cycle arrest time after the challenge dose by quantification of the mitotic index and direct EdU incorporation based replication assays. The conclusion was that RAR-induced flies recover faster and hence likely suffered less damage or repaired the damage more efficiently.

Working with a genetic model organism, the authors employed a panel of DNA repair mutant flies and this revealed a dependence of the RAR-protective effect on a fully functional DNA repair system. Furthermore, fewer gamma-H2Av foci were detected after the challenge dose in RAR-induced cells; consistently, an up-regulation of Rad50 and Nbs1 was observed on the protein level. Since the RAR appears to increase the efficiency of HR-mediated DNA repair, the authors tested whether telomeric fusions are also affected. Hence, instead of a challenge dose telomere uncapping mutations were analyzed in RAR induced and untreated flies. RAR could indeed also alleviate this problem, again in a manner that depends on an intact DNA repair pathway. Further mechanistic insight was gained by gene expression analysis (RNA seq). Specifically, the comparison of expression levels after the challenge dose only vs. the challenge dose in RAR-induced animals has the potential to reveal genes that mediate the RAR effect. Interestingly, this

comparison uncovered expression changes in RNA processing genes, including a reduced expression of the D-isoform of Loqs. qRT-PCR analysis corroborated the RNA-seq finding, although the change was marginal yet significant. The genetic studies, however, revealed that the RAR can induce changes in siRNA generation that are qualitatively similar to a loss of Loqs. Furthermore, in loqs mutants the number of chromosome breaks after the challenge dose was reduced (and dependent on ATM), this could be rescued by re-expression of the D-isoform and RAR-induction did not further lower the number of chromosome breaks in loqs mutant animals. Finally, Loqs-PD can reduce the number of telomere fusions in decapping mutants.

This manuscript reveals a very interesting and so far undescribed link between the D-isoform of Loqs and DNA repair. The experiments are well described and although I find some observations rather puzzling (like the trans-generational inheritance), I cannot see any obvious faults in the experimental design and analysis. I do want to say, however, that I am not an expert on irradiation treatment and the cytological analysis of chromosome breaks. Given that there is very little mechanistic information about the RAR, this is an important progress. However, the Loqs-D isoform is likely an insect-specific variant and hence it is currently unclear whether this can be extended to vertebrates.

I mainly want to say a few words to the discussion of the findings:

The authors should be more careful in their discussion about the implication of DNA-damage induced small RNAs. The genetic identification of Loqs-D does not necessarily imply that this is via its known role in the biogenesis of siRNAs. The damage-induced small RNAs are clearly present in *Drosophila*, but a study has shown that Dcr-2 and Ago2 are not required for homology directed repair. This was in a very different assay system not involving irradiation or telomere uncapping, but it is nonetheless quite possible that the RAR-effect of a Loqs-mutation is not via small RNAs. The authors cite this study, yet they still conclude that the significance of siRNA formation at DNA breaks in *Drosophila* "remains unaddressed". Clearly, the published study does not address all possible questions, in particular concerning repair after irradiation, but it does at least address the question nonetheless.

Furthermore, while there is now ample evidence that transcription initiates at DNA breaks (and likely telomeres as well), the original descriptions of damage-induced SMALL RNAs in human cells and plants are also now considered somewhat problematic. The plant study solely relied on transgenes and the effect of a Dcr knockdown on the DR-GFP reporter can be rescued by re-introduction of the miRNA let-7 (Liu et al, NAR 2015, doi: 10.1093/nar/gku1368), a "Dicer and Drosha dependent" miRNA. The Francia et al study describing the role of Dicer and Drosha dependent small RNAs (ddRNAs) at DNA breaks has some serious flaws: First, treating permeabilized cells with RNase will have massive consequences on nuclear and chromatin structure that go far beyond small RNAs, and this was a known fact for decades. Second, the authors describe an experiment where the proposed DNA repair defect could be rescued with the small RNA fraction isolated from independently irradiated cells. To me, this is proof that the RNAs can NOT be locus-specific, as it would require radiation-induced DNA breaks at the same sites in independent cells (or else shredding the genome to rather small pieces with the irradiation). Finally, the small RNA sequencing data they provide from a cleavable locus shows only background levels of siRNAs, so low that I consider this proof for the ABSENCE of small RNAs. The field also has not really produced many follow-up studies that substantiate the claim of small RNA requirement for DNA repair. All this is of course my personal view and I certainly do not request the authors to re-iterate this in their discussion, but I do want to suggest that these papers and their claim of small RNAs without reflecting on their critical issues.

The discussion of esiRNAs falls a bit short of its potential – Loqs-D is clearly required for the production of these long hairpin-derived small RNAs, which have a biogenesis similar to siRNAs yet compare to miRNAs in their abundance and potentially mode of action. They seem to preferentially target mitochondrial factors and RNAi-mutants (Dcr-2, Ago2 and also Loqs to some extent) show impaired spermatogenesis and male sterility. (e.g. Papers from the Lai-lab: DOI: 10.1016/j.molcel.2014.11.025, doi: 10.1016/j.devcel.2018.07.004). Although I do not see a direct connection from this to radiation-induced chromosome breaks, the mitochondria and perhaps their ROS-generating activity may contribute to the adaptive response. In other words, it is conceivable that Loqs acts in part via modulation of mitochondrial activity. I invite the authors to speculate about this, in addition to the locus-specific small RNAs generated at DNA breaks.

Reviewer #3 (Remarks to the Author):

In this manuscript by Parrazzo and colleagues, the authors seek to determine whether low-dose exposure to ionizing radiation (LDR; "priming") elicits a protective effect on genome stability following a high-dose exposure to ionizing radiation (IR). Due to conflicting results in mammalian cells, *Drosophila* has been used as a model system. The authors found that LDR (0.4Gy) applied to embryos and larvae of wild-type flies indeed results in a protective effect on genome stability after high dose IR (10Gy), as measured by chromosomal breaks, telomere fusions after genetic uncapping, faster recovery from G2/M checkpoint, and decreased gH2Av formation. They also demonstrated that the chromosome breakage and telomere fusing is dependent on a variety of DNA damage response and DNA repair genes. They also investigated the effect of IR priming followed by high dose exposure on gene expression, and identified a decrease in expression of the D isoform of Loquacious, a well conserved dsRNA binding protein required for the esiRNA biogenesis through its interaction with Dicer-2.

This manuscript demonstrates novel findings that addresses the conflicting data regarding the protective effect of low dose IR exposure by providing a mechanism for this phenomenon. It would be of interest to those in the DNA repair field, particularly those interested in the impacts of low dose irradiation. The work also further validates the use of model organisms to address discrepancies in the literature and/or fail to study the effects in multi-cellular systems.

Overall, the rationale for the experiments is well demonstrated and the experiments themselves are thorough. For example, the researchers demonstrate that low dose alone does not affect survival or fertility (Fig. S5); they distinguish the mechanism of telomere fusion between uncapping and telomere elongation (Fig. S4); and genetically (and creatively) demonstrate a decrease in RNAi capacity after LDR due to the decrease expression of Loq-D (Fig. 6b and 6c). For the most part, results are fairly interpreted, and the discussion covers the major points. Methods and figures are presented clearly and the manuscript is very well-written. I only have a few minor points to address to improve the manuscript:

1. For the chromosome aberration experiments, embryos and larvae were pre-treated with low dose IR. A high dose treatment was then applied and cells examined 4 and 8h later in order to measure chromosome integrity after high-dose IR induced damage in cells in G1 (8h) and S-G2 (4h). How are the authors sure that the differences observed (i.e., no protective effect in 8h treatment group) is indeed due to cell cycle phase, and not a timing issue, where the additional 4 hours provide time for the cell to repair the damage?
2. For RNA-seq analysis, why were male brains used in sample collection?
3. In general, for statistical analyses, two groups were always compared. However, it is not clear in the figures which two groups. Could the authors please include this description in the figure legends, particularly for Figures 1b and 6d. Related, is it safe to assume that any data points that do not have a p annotation are presumably n.s.?
4. Could you explain the source of the BlmD3 mutant allele (it is not mentioned in the Materials and Methods)? To my knowledge, the available stock is homozygous lethal due to secondary site mutations.
5. For WB analysis, Line 639 states that the data comes from three independent western blots. Were the protein lysates used in each Western blot also from three independent experiments, or the same lysates used for three different blots? Please clarify.
6. In Figure 5, it is unclear with the "B", "D" and "C" labels come from and it is challenging to follow. Could another label be used that is more intuitive to the reader? For example for B vs. D,

you could label as "0.4+10 vs. 10". Regardless of whether the authors want to change the labels as suggested, each label should be defined in the figure legend.

7.Minor grammatical edits:

a.Line 223 "DSB" should be "DSBs"

b.Line 269 "has not a" should be "does not have a"

c.Line 502, formatting of "micro" is necessary

d.Line 636, "LDDR" should be "LDR"

e.Line 637, "wild type females" should be "wild-type females"

f.Various references need appropriate capitalization and journal title abbreviations (i.e., line 888, Human & experimental toxicology should be Hum Exp Toxicol, etc.)

Point-by-Point Response to Reviewers

Reviewer #1

(Remarks to the Author):

Comments for Porrazzo et al.

In Radio Adaptive Response (RAR), an initial exposure of a low dose radiation protects the organism from a subsequent challenge of radiation at a much higher dosage. Although RAR has been known for a long time, the underlying mechanism remains obscure. In this study, the authors investigated the genetic control of RAR using the classic Drosophila model. Although most of the studies remains descriptive, the amount of cytological work, the number of mutations tested, and the variety of genetic pathways covered in this study are unprecedented. In addition, the authors' uncovering of Loquacious, a small RNA biogenesis factor, as a potential modulator of RAR adds new directions to future studies of RAR.

We thank this reviewer for her/his positive comments on our results

I only have minor comments for the authors.

(1) Tables in Figures 1, 2, and 6: it would be helpful to state specifically the pair-wise comparison from which a p value was generated for as there are so many different combinations possible.

We agreed with this reviewer concern and modified the Legends of Figures 1, 2 and 6 accordingly

Fig 1(b) Analysis of the frequency of CBs induced by the acute exposure to 10Gy, in 0.4Gy LDR chronically treated and non-treated (NT) larvae after 4 and 8h post irradiation (PIR), and in the first and the second generation from adults exposed to 0.4Gy LDR during their development. Chromosome exchanges did not vary and were not considered in the analysis. At least three independent experiments were conducted. Error bars represent the standard errors of the mean. LDR: low dose rate; ISOs: isochromosome deletions; CDs: chromatid deletions; CBs: chromosome breaks; PIR: post-irradiation time; F1: first filial generation; F2: second filial generation. **A comparison between data for (0.4 Gy + 10 Gy) and 10 Gy, for F1 and 10 Gy and for F2 and 10 Gy was performed using Student's t-test; †: statistically significant for p<0.05, ††: statistically significant for p<0.01**

Fig 2(e) DNA repair gene mutant larvae do not elicit RAR. Frequency of CBs induced by an acute 2Gy, 5Gy or 10Gy irradiation on 0.4Gy LDR Drosophila strains mutated in genes involved in the DNA damage response (DDR). The ATPCL mutant was used as a control. Note that in all DNA repair (but not in 23 ATPCL) mutants RAR is abolished. ISOs: isochromosome deletions; CDs: chromatid deletions; CBs: chromosome breaks; NT: non-treated; PIR: post-irradiation time. **For each mutant a comparison between data for priming + challenging dose and challenging dose alone was made using Student's t-test; †: statistically significant for p<0.01. Bar= 10µm.**

Fig 6(d) Frequency of IR-induced CBs in 0.4Gy LDR treated and non-treated (NT) homozygous and heterozygous LoqsKO mutants as well as OR-R controls, in loqsKO null mutants expressing the Loqs-PD and Loqs-PB exogenous isoforms. CBs: chromosome breaks. **A comparison between data for Loqs mutants (0.4 +10 Gy) and OR-R (10 Gy) and for Loqs mutants (10 Gy) and OR-R (10 Gy) was performed using Student's t-test; †: statistically significant**

(2) Page 3, line 61, please define "LET".

We added the definition of LET (**Linear Energy Transfer**) in the Introduction

(3) It would be helpful to further describe the difference between Loq-D and the other isoforms in terms of domain structures.

On page 11 We have added the following additional details on the differences among all 4 Loqs isoforms:

Loqs PA and Loqs PB both harbor three dsRBDs (L1, L2L3) while Loqs PC and PD both lack the third dsRBD and instead have short aa stretches at their C-termini

(4) Please make sure all comparisons are supported by robust statistics.

We have indeed compared the means of pairs of datasets, each obtained from at least three independent experiments. We believe that all comparisons are supported by fairly solid statistics.

Reviewer #2

....This manuscript reveals a very interesting and so far undescribed link between the D-isoform of Loqs and DNA repair. The experiments are well described and although I find some observations rather puzzling (like the trans-generational inheritance), I cannot see any obvious faults in the experimental design and analysis. I do want to say, however, that I am not an expert on irradiation treatment and the cytological analysis of chromosome breaks. Given that there is very little mechanistic information about the RAR, this is an important progress. However, the Loqs-D isoform is likely an insect-specific variant and hence it is currently unclear whether this can be extended to vertebrates...

We have really appreciated the positive feedback provided by this reviewer on our manuscript. We also quite agree with the reviewer general comments that careful attention should be paid on the complicated issue that deals with the involvement of sRNAs in the DNA damage response. In our manuscript we had indeed discussed the evidence supporting this reviewer statement that smRNAs are not per se involved in the DNA damage response in *Drosophila*, ruling out the involvement of Loqs-mediated siRNA biogenesis in LDR protective effects. To further meet this reviewer request of not emphasize any role for siRNAs in LDR we removed our data (and the corresponding figures 6 b,c) on the GMR-wIR flies that we understand could mis-lead to an overintepretation of the role of esi-biogenesis in the LDR protective effect.

Although I do not see a direct connection from this to radiation-induced chromosome breaks, the mitochondria and perhaps their ROS-generating activity may contribute to the adaptive response. In other words, it is conceivable that Loqs acts in part via modulation of mitochondrial activity. I invite the authors to speculate about this, in addition to the locus-specific small RNAs generated at DNA breaks

Following this reviewer suggestion, we checked the levels of reactive oxygen species (ROS) in unirradiated, 10Gy- (B), and 0.4Gy LDR + 10Gy-irradiated (D) Oregon R larval brains as well as in larval brains from both unirradiated and 10Gy treated *Loqs* mutants using dihydroethidium (DHE) as an indicator. We found that ROS levels in 10Gy-irradiated brains, were undistinguishable from those observed in 0.4Gy LDR + 10Gy-irradiated brains, although, as expected, they increased compared to unirradiated larvae. Moreover, we have also found that ROS levels of 10Gy treated *Loqs* mutants were similar to that found in either 10Gy- or 0.4Gy LDR + 10Gy-irradiated controls. These results indicate that the LDR treatment and the deprivation of *Loqs* do not lead to a statistically significant response to the ROS generating activity that could be considered to explain the radioresistance to IR-induced chromosome breaks. We have integrated these observations in the text (providing also Figures in the Supplementary Material) as follows:

On page 7.

...Moreover, using dihydroethidium (DHE) as an indicator we found that ROS levels in 10Gy-irradiated brains did not differ from those observed in 0.4Gy LDR + 10Gy-irradiated brains, although, as expected, they increased compared to unirradiated larvae (Supplementary Figure 4). These results indicate that LDR does not lead to a statistically significant response to the ROS generating activity that could be taken into account to explain the radioresistance.

On page 12.

Finally, we observed that the kinetics of γ H2Av recruitment and the response to production of

ROS in *loqs*^{KO} and in 0.4Gy LDR + 10Gy cells, were similar (Figure 6e; Supplementary Figure 4).

Reviewer #3 (Remarks to the Author):

This manuscript demonstrates novel findings that addresses the conflicting data regarding the protective effect of low dose IR exposure by providing a mechanism for this phenomenon. It would be of interest to those in the DNA repair field, particularly those interested in the impacts of low dose irradiation. The work also further validates the use of model organisms to address discrepancies in the literature and/or fail to study the effects in multi-cellular systems.

We are grateful to this reviewer for her/his appreciation in our data

1. For the chromosome aberration experiments, embryos and larvae were pre-treated with low dose IR. A high dose treatment was then applied and cells examined 4 and 8h later in order to measure chromosome integrity after high-dose IR induced damage in cells in G1 (8h) and S-G2 (4h). How are the authors sure that the differences observed (i.e., no protective effect in 8h treatment group) is indeed due to cell cycle phase, and not a timing issue, where the additional 4 hours provide time for the cell to repair the damage?

It has been previously demonstrated that *Drosophila* mitotic G1 and G2 cells take 8-10 hours and 4 hours to enter mitosis, respectively (see for example Gatti et al., *Genetics* (1974) **77**: 701-719). Thus, it is normally accepted that 8hrs and 4hrs PIRs after IR-treatment allow the evaluation of the effects of IR on chromatin before and after S phase. IR-induced CB frequency after 8hr PIR is therefore lower than that after 4hr PIR as G1 cells have more time than G2 cells to repair DNA breaks. We have indeed found that IR-induced CBs frequency is reduced in LDR-treated cells only after 4hr (but not 8hr) PIR indicating that the LDR exerts a protective effect mainly after the S phase.

2. For RNA-seq analysis, why were male brains used in sample collection?

To address this reviewer criticism, we have inserted the following text on page 10.

We decided to focus our transcriptomic analyses on males to avoid the confounding effects of gene expression variability of females which is known to be affected by dietary conditions during development and related to investment into reproduction-related processes (May et al., 2021; Ayroles et al., 2009).

3. In general, for statistical analyses, two groups were always compared. However, it is not clear in the figures which two groups. Could the authors please include this description in the figure legends, particularly for Figures 1b and 6d. Related, is it safe to assume that any data points that do not have a p annotation are presumably n.s.?

See response to criticism N.1 or Reviewer #1

4. Could you explain the source of the BlmD3 mutant allele (it is not mentioned in the Materials and Methods)? To my knowledge, the available stock is homozygous lethal due to secondary site mutations.

We received the BlmD3 allele (previously named as *mus309*) from Prof. M. Gatti (Sapienza University of Rome; we have provided this information in the Materials and Methods.) who obtained it from Prof. Green (Boyd et al., 1981; *Genetics* 97: 607-623) long time ago. The D3 mutation was confirmed by sequencing (Piergentili R., *Dros. Inf. Serv.* 97 (2014). The BlmD3 mutant allele was originally female sterile but has subsequently acquired a secondary site late-lethal mutation. However, homozygous larvae are rare (as indicated by the limited number of cells examined).

5. For WB analysis, Line 639 states that the data comes from three independent western blots. Were the protein lysates used in each Western blot also from three independent experiments, or the same lysates used for three different blots? Please clarify.

We clarified this point in the Legend of Figure 3 as follows: **WBs from three independent experiments were used for the quantification**

6. In Figure 5, it is unclear with the “B”, “D” and “C” labels come from and it is challenging to follow. Could another label be used that is more intuitive to the reader? For example for B vs. D, you could label as “0.4+10 vs. 10”. Regardless of whether the authors want to change the labels as suggested, each label should be defined in the figure legend.

We agreed with this reviewer suggestion and changed the labels and legends of Figure 5 accordingly as follows:

... The analysis was performed on three biological replicates obtained from three independent experiments, namely **1, 2, and 3 from the 10Gy (B) and 0.4Gy LDR + 10Gy (D) conditions.**

7. *Minor grammatical edits:*

a. Line 223 “DSB” should be “DSBs”

b. Line 269 “has not a” should be “does not have a”

c. Line 502, formatting of “micro” is necessary

d. Line 636, “LDDR” should be “LDR”

e. Line 637, “wild type females” should be “wild-type females”

f. Various references need appropriate capitalization and journal title abbreviations (i.e., line 888, Human & experimental toxicology should be Hum Exp Toxicol, etc.)

We have fixed all typos indicated (see corrections in red in the text)

REVIEWERS' COMMENTS:

Reviewer #2 (Remarks to the Author):

First of all I would like to apologize for the delayed response. The authors have addressed my concerns meticulously, the ROS assays are indeed an appropriate experimental approach - absence of evidence is not evidence of absence, however, so the fact that no changes are detected in the Loqs mutant does not fully rule out that this is indirect. I do think that the readers are aware of this, though, and the detected increase upon irradiation does give a good indication of the sensitivity of the assay. Altogether I think, as stated above, that this is an appropriate experiment to address the concern.

I still find that the sentence in the discussion on Loqs and DNA repair is wrong (lines 429-431, p. 14). First, the inactivation of the siRNA pathway does not lead to HOMOLOGY-MEDIATED DNA repair defects (only those were tested in the paper). Second, the significance of siRNAs in DNA repair HAS been addressed and, as of the current status of the published literature, they play no role for homology-mediated repair. Of course this paper has its own limitations and there are certainly possibilities why a role of siRNAs has remained undetected - but not "unaddressed". Thus, one should change "unaddressed" into "unclear" in that sentence.